# Research Progress on Starfish Outbreaks and Their Prevention and Utilization: Lessons from Northern China

**DOI:** 10.3390/biology13070537

**Published:** 2024-07-17

**Authors:** Liang Qu, Yongxin Sun, Chong Zhao, Maurice R. Elphick, Qingzhi Wang

**Affiliations:** 1Dalian Key Laboratory of Genetic Resources for Marine Shellfish, Liaoning Ocean and Fisheries Science Research Institute, Dalian 116023, China; quliang085@163.com (L.Q.); sunyongxin1977@163.com (Y.S.); 2Key Laboratory of Protection and Utilization of Aquatic Germplasm Resource, Ministry of Agriculture and Rural Affairs, Dalian 116023, China; 3Dalian Jinshiwan Laboratory, Dalian 116034, China; 4Key Laboratory of Mariculture & Stock Enhancement in North China’s Sea, Ministry of Agriculture and Rural Affairs, Dalian Ocean University, Dalian 116023, China; chongzhao@dlou.edu.cn; 5School of Biological and Behavioural Sciences, Queen Mary University of London, London E1 4NS, UK

**Keywords:** starfish, *Asterias amurensis*, outbreak, prevention, utilization

## Abstract

**Simple Summary:**

Starfish are the natural enemies of shellfish. In many areas, especially in Qingdao, China, shellfish aquaculture has been adversely affected by starfish predation for many years, resulting in serious economic losses. To facilitate strategies for the protection of shellfish from starfish, this review analyzes the reasons for the proliferation of starfish and proposes governance suggestions for the control and utilization of starfish.

**Abstract:**

Starfish are keystone species as predators in benthic ecosystems, but when population outbreaks occur, this can have devastating consequences ecologically. Furthermore, starfish outbreaks and invasions can have adverse impact economically by impacting shellfish aquaculture. For example, an infestation of starfish in Qingdao led to a 50% reduction in sea cucumber production and an 80% reduction in scallop production, resulting in an economic loss of approximately RMB 100 million to oyster and other shellfish industries. Addressing the imperative need to proactively mitigate starfish invasions requires comprehensive research on their behavior and the underlying mechanisms of outbreaks. This review scrutinizes the historical patterns of outbreaks among diverse starfish species across various regions, delineates the factors contributing to the proliferation of *Asterias amurensis* in Chinese waters, articulates preventive and remedial strategies, and outlines the potential for the sustainable utilization of starfish.

## 1. Introduction

Starfish, belonging to the phylum Echinodermata and the class Asteroidea, exhibit a planktonic lifestyle during their larval stage and transition to a benthic lifestyle in adulthood [1]. They are globally distributed, with several orders, including Paxillosida, Forcipulatida, Valvatida, and Spinulosa, found in seas/oceans around China. Common species include *Luidia quinaria* von Martens, 1865; *Asterias amurensis*; *Patiria pectinifera*; and *Solaster dawsoni* [2].

As carnivores, they trigger a trophic cascade by reducing herbivore grazing, indirectly benefiting primary producers [3]. However, when herbivores overpopulate and harm the ecosystem via overgrazing, predator reintroduction may not have the expected cascade effect [4,5]. Starfish have a particular predilection for shellfish, primarily feeding on clams, scallops, oysters, and abalones when they congregate in large numbers [6]. This predation significantly impacts the diversity of marine benthic organisms and presents substantial challenges to bottom seeding in shellfish breeding areas [7]. As China is a country with a large aquaculture industry, frequent outbreaks of starfish have seriously harmed the economic interests of shellfish aquaculture. In China, the starfish species *A. amurensis* and *P. pectinifera* (formerly *Asterina pectinifera*) pose the most significant threats to aquaculture. These starfish species utilize their tube feet to open gaps in shells and subsequently evert their cardiac stomach to envelop the soft tissues of their prey. They also secrete digestive fluids externally to digest and consume the softer parts of their prey [8]. The tube feet of starfish operate by discharging adhesive proteins [9], which prevents prey from escaping once ensnared. Consequently, starfish pose a significant threat to aquaculture, particularly to juvenile shellfish.

In the realm of starfish research, the extant literature includes topics such as behavior, feeding habits, developmental biology, and neurobiology [10,11,12,13]. However, the exploration of starfish outbreak mechanisms and preventive strategies is still in its infancy, marked by a dearth of studies addressing biological prevention and control methodologies, along with resource utilization. This article undertakes a comprehensive review of the worldwide prevalence of starfish outbreaks, encapsulates existing management and utilization approaches for starfish on both national and international scales, and proffers recommendations aimed at mitigating the adverse impact of starfish outbreaks. The overarching goal is to safeguard marine ecosystems and curtail economic losses in aquaculture areas.

## 2. Starfish Outbreak Disasters and Their Causes

### 2.1. Recent Starfish Outbreaks in Various Regions

Starfish outbreaks have been a recurring phenomenon in various regions since the 1950s, with the first recorded instance in Tokyo Bay, Japan [14,15]. These outbreaks have led to significant mortality among shellfish and juvenile gastropods on the seabed, causing profound changes in the ecosystem composition of benthic communities [16,17,18]. In 2006–2007, a starfish outbreak in Qingdao, China, resulted in substantial economic losses for abalone and clam farming operations within the affected waters [6]. Similarly, in Jiaozhou Bay in 2007, the density of starfish reached as high as 300 individuals per square meter, nearly decimating local clam and abalone fry [6]. Up to 60% of the clam area, primarily inhabited by *Ruditapes philippinarum*, was infested by starfish, resulting in economic damage amounting to tens of millions of RMB [19].

An infestation of *P. pectinifera* led to a 50% reduction in sea cucumber (*Apostichopus japonicus*) production in Weihai aquaculture areas, and also in 2007 [20]. Starfish, predominantly of the *Asterias* genus, were widespread in scallop (*Azumapecten farreri*) breeding cages in Liuqing River, resulting in a mortality rate of up to 80% [8]. Subsequently, starfish continued to pose a threat to local shellfish. In 2012, an *A. amurensis* infestation in Qingdao’s scallop farming areas resulted in damage exceeding 80% of the scallop production for that year. Between 2020 and 2023, Jiaozhou Bay witnessed recurring and massive starfish invasions, with the invasion season advancing annually. The starfish outbreak in Qingdao in 2021 alone incurred an economic loss of approximately RMB 100 million to oyster and other shellfish industries [21].

The starfish outbreak that erupted in Jiaozhou Bay in 2021 is shown in Figure 1. In addition to *A. amurensis*, the pervasive invasion of *P. pectinifera* also exerts a negative impact on numerous aquaculture areas. In regions dedicated to shellfish farming, *P. pectinifera* demonstrates a remarkable ability to locate buried shellfish by scent and feeds on them extensively. Observations have noted the presence of *P. pectinifera* in shellfish release areas, with their aggregation density increasing in correlation with the release density [12]. Beyond shellfish, *P. pectinifera* poses a significant threat to sea cucumbers and competes for food with other benthic organisms [22]. 

Starfish outbreaks not only inflict direct harm on shellfish but also pose a substantial threat to corals. The crown-of-thorns starfish (*Acanthaster planci*, COTS) is recognized as the primary natural predator of corals, and repeated COTS outbreaks have led to the degradation of Indo-Pacific coral reefs [23]. The damage caused is comparable to that resulting from coral bleaching due to climate warming [24]. Since the 1960s, the outbreak cycle of COTS has been progressively shortening [25]. This coupled with deteriorating climate and environmental conditions has amplified the challenges associated with coral reef restoration, raising concerns about the potential for full recovery [26]. 

The map in Figure 2 shows outbreaks of starfish species in various regions of the world, and Table 1 provides a summary of recent starfish outbreak disasters and their impacts in various regions.

### 2.2. Reasons for Starfish Outbreaks

Extensive research into starfish outbreaks in northern China has indicated that large-scale incidents are intricately linked to climate variations, environmental changes, the overexploitation of their natural predators, and the availability of essential food sources [32]. The “increased planktonic prey” hypothesis suggests that human-induced seawater eutrophication, propelled by increased anthropogenic activities, results in extensive planktonic algal blooms. These blooms subsequently provide a substantial food source for starfish planktonic larvae, facilitating their survival in significant numbers [32]. The “declining predator” hypothesis proposes that adult starfish face minimal threats from natural predators. A marked reduction in the population of overfished carnivorous fish, which typically prey on starfish larvae, has enhanced the survival rate of the larvae. While the definitive cause of large-scale starfish outbreaks remains unconfirmed [32], the frequency of population surges has risen in tandem with improved larval survival rates.

Upon conducting a comprehensive literature review, we have identified three critical factors that contribute to starfish outbreaks:

#### 2.2.1. Robust Adaptability of Starfish

Starfish exhibit extraordinary adaptability across a broad spectrum of environmental conditions, thriving in habitats that span from intertidal zones to depths reaching 40 m [33]. They demonstrate an impressive tolerance to fluctuations in temperature and salinity, with the latter ranging from 12 to 36 degrees centigrade. Their global distribution is well documented, with their presence reported in waters around the world [14,28,34,35,36,37,38].

Despite their relatively brief lifespan, averaging around three years [39], starfish display significant reproductive capabilities. An adult female starfish can lay an astonishing quantity of eggs simultaneously, with estimates reaching up to 100,000 [34]. Moreover, starfish have remarkable regenerative abilities. When confronted with threats or survival challenges within their environment, they may resort to self-amputation (autotomy) as a strategy for escape and self-preservation. Over time, a new arm regenerates at the site of autotomy, and in some starfish species, the severed arm can develop into a completely new starfish [40].

#### 2.2.2. Abundant Food Resources

During their planktonic larval stage, starfish primarily consume unicellular algae found among the phytoplankton. Increased human activities have led to nutrient runoff from land that infiltrates the seas, resulting in rapid phytoplankton blooms and consequently, increased survival rates of starfish larvae [32]. Adult starfish predominantly prey on shellfish and are formidable feeders. The abundance of neural sensory cells in their epidermis enables them to detect prey with precision [6], responding swiftly to external stimuli. While starfish typically exhibit sluggish locomotor activity, their speed increases significantly when prey is detected. Thus, in aquaculture regions with substantial shellfish production, large populations of starfish can be attracted for a feeding frenzy (e.g., *A. amurensis* in Qingdao’s Jiaozhou Bay). Furthermore, the phenomenon of “cannibalism” exists among starfish, where smaller, less vigorous individuals are either devoured by their stronger counterparts or have their limbs amputated. Over time, this phenomenon eliminates weaker starfish, intensifying shellfish predation in areas with starfish outbreaks [22,27].

#### 2.2.3. Decline in Natural Predators under the Influence of Climate Change

Triton’s trumpet (*Charonia tritonis*), recognized as the natural adversary of COTS [38], has unfortunately become a rare sight due to factors such as escalating global temperatures and environmental degradation. Adult starfish are virtually impervious to predation by other organisms due to their robust exoskeletons, a key factor contributing to their widespread proliferation and resilience. Starfish reproduction involves external fertilization, with adult starfish releasing sperm and eggs into the seawater. The resulting larvae undergo a planktonic phase, during which they are primarily vulnerable to predation. Fish species, such as *Larimichthys polyactis* and *Lateolabrax japonicus*, are known to prey on starfish larvae [20]. In recent years, climate-related factors have led to a decline in the population of fish and seabirds in coastal regions that prey on starfish. This decline has resulted in reduction in predation rates on the prey and an increased survival rate among starfish larvae [31], thereby concealing a latent threat to the potential outbreak of starfish populations.

#### 2.2.4. Summary of the Reasons for Starfish Outbreaks

In conclusion, these factors provide crucial insights into the mechanisms driving starfish outbreaks and further research and management strategies are essential to address the complex dynamics contributing to these phenomena.

### 2.3. Control and Prevention of Starfish Outbreaks

Given the biological and physiological characteristics of starfish, researchers both domestically and internationally have conducted comprehensive studies and initiatives to control starfish from various angles, yielding significant practical results. This review presents a compilation of starfish removal strategies, categorized into physical, chemical, and biological approaches, for a more comprehensive understanding. Control and prevention methods is shown in Table 2.

#### 2.3.1. Physical Removal Methods

Manual timed cleaning is one of the simplest, most effective, and inexpensive methods to rapidly control the spread of starfish in a specific area [10]. Since 2006, the city of Qingdao has been using artificial fishing to remove the proliferating *A. amurensis*. The conventional approach for eliminating starfish from artificial marine breeding zones and areas dedicated to shellfish bottom sowing and breeding often entails concentrated destruction by divers, a process that can prove to be challenging and inefficient. Fishing boats can catch between 1500 and 2500 starfish at a time, with a maximum daily catch of nearly 30,000. For the fishing of starfish, it is crucial to properly plan the timing to achieve maximum results with minimum effort, and it is generally more effective to do so before the starfish enter their breeding season. For instance, in Qingdao, the breeding season of *A. amurensis* typically spans from February to September [10].

The feeding rate of *A. amurensis* is closely related to the temperature [14,34]. Within the temperature range of 5–15 °C, the feeding activity of *A. amurensis* intensifies with rising water temperatures. In the surrounding waters of Qingdao City, the water temperature typically reaches the optimal range for *A. amurensis* during May to June [10], coinciding with the peak of their predation. This period provides a strategic window for conducting fishing operations to enhance precautionary measures. Subsequently, the feeding rate of *A. amurensis* decreases with the increase in water temperature. By August, the feeding rate of *A. amurensis* reduces considerably, marking it as the most opportune time for extensive removal. This strategy effectively prevents *A. amurensis* from engaging in substantial feeding after the water temperature decreases, which is a preparation for their reproduction. As the water temperature rises again, the feeding rate of *A. amurensis* diminishes. By August, this reduction becomes significant, making it the ideal time for mass removal.

This physical removal method relies mainly on fishers casting nets to drag the starfish. This method may also catch a small amount of seaweed and benthic shellfish, but most are empty shells that have been eaten by the starfish.

#### 2.3.2. Chemical Removal Methods

The use of chemical agents emerges as an effective method for starfish removal, operating on the premise that it efficiently eradicates starfish while ensuring the protection of economically significant species within the cultured area. The commonly used method in the early stages involved spreading quicklime to repel starfish: spreading quicklime on the seedling collector of the Yesso scallop (*A. farreri*), or directly immersing the seedling collector in a quicklime solution, with a recommended low concentration (0.1%) of quicklime and requiring a prolonged soaking time. This method has achieved satisfactory results in the Yesso scallop farming areas [27].

Research by Li et al. (2014) [10] identified a pronounced aversive response of *A. amurensis* to high concentrations (1–3 mol/L) of ammonium salt and acetic acid stimulation. Leveraging this observed behavior, the deployment of slow-release ammonium salt pouches in the aquaculture zone prior to seeding emerged as a strategic means to effectuate the dispersion of *A. amurensis*. However, due to uncertainties surrounding the potential adverse environmental impacts of these chemicals on marine ecosystems and other aquatic organisms, direct spraying of these substances into seawater is not recommended. To mitigate the risk of marine environment contamination, an alternative approach may involve precision injections, a technique reminiscent of the Australian model, wherein submersible robots are trained to identify COTS and administer acid-based pharmaceuticals [35]. This method not only eliminates the need for manual labor but also safeguards the environment while achieving large-scale eradication objectives.

The starfish central nervous system comprises radial nerve cords and a circumoral nerve ring, whilst sensory organs (including chemosensory organs) are concentrated at the tips of the arms and these enable starfish to perceive chemical signals from the external environment [36]. These chemical cues, often originating from potential prey, play a crucial role in attracting starfish for feeding, making them susceptible to targeted trapping using chemical signals. Notably, research has revealed that 5,8,11,14-eicosatetraenoic acid, also known as arachidonic acid, coupled with α-linolenic acid, represents essential unsaturated fatty acids that COTS cannot synthesize autonomously. Instead, they rely on a consistent dietary source from corals rich in these fatty acids [37]. This dietary dependence serves as a key contributor to the recurrent outbreaks of *Acanthaster* in coral regions, suggesting that these fatty acids hold promise as attractants for starfish capture. In a practical application, Japanese researchers conducted sea-based experiments, successfully capturing seven starfish within a 1.5-square-meter trap over a nine-day period using α-linolenic acid as an attractant [39]. This outcome holds potential significance for future research endeavors aimed at protecting coral reefs from the predatory impact of starfish.

Aquatic organisms rely on specialized receptors to perceive chemical signals in their external milieu, a capability fundamental to their normal life processes. In the realm of marine invertebrates, reproduction involves the dispersion of eggs and sperm into the aquatic environment. The success of fertilization is intricately linked to the synchronous release of gametes [40,41]. Studies have illuminated the mechanisms underpinning the reproductive behavior of male and female COTS, revealing their capacity to orchestrate reproductive aggregation and synchronized spawning through the reception of sex-specific pheromones via their sensory tentacles [42]. Jönsson’s [43] research findings further delineate this phenomenon, elucidating the regulation of aggregation among COTS in the lead-up to spawning. This regulation is facilitated through the release of pheromones, generating a complex chemical signal comprising hundreds of proteins. Among these identified proteins, 84 have been identified in wild COTS, with 36 demonstrating significant differential expression between genders. This knowledge underpins the notion of utilizing differential metabolites derived from male and female starfish as potential sources of information for behavioral studies, aimed at discerning the presence of active pheromones and exploring the principal constituents of various starfish pheromones. Practically applying this understanding, ground cages have been strategically deployed in the peripheral waters of aquaculture regions. These cages incorporate attractant dispensers releasing starfish sex pheromones, effectively capitalizing on the inherent attraction of starfish to these chemical cues. Consequently, when lured in significant numbers, the starfish can be systematically removed, thereby contributing to effective population control measures.

#### 2.3.3. Removal Methods through Natural Enemy Deployment: Biological Control

A robust exoskeletal structure adorned with spines renders starfish relatively impervious to predation in their adult phase. However, as highlighted above, an internationally recognized natural adversary of COTS is *Triton’s trumpet*. Furthermore, remnants of *A. amurensis* have been uncovered in southeastern Tasmania’s Derwent estuary, hinting at the presence of predators [38]. Detailed observations have revealed that the large crustacean known as the spider crab (*Leptomithrax gaimadii*) is a voracious consumer of *A. amurensis*. Furthermore, in controlled settings within aquaria in Japan and the United States, instances of Japanese solar starfish (*Solaster paxillatus*) and red emperor crabs (*Paralithodes camtschaticus*) preying upon *A. amurensis* have been documented [10].

The herring gull *Larus argentatus* Pont feeds on adult starfish by grabbing and shaking their arms to break them [44], and during the larval phase of starfish development, various organisms, including *L. polyactis*, *L. japonicus*, and *Acanthopayrus schlegelii* [32], actively feed on them. Hence, the strategic placement of natural predators is most effective when executed before starfish reach developmental maturity [27]. Carnivorous fish can be introduced to the aquaculture region during the starfish’s reproductive phase, specifically to target the gametes, larvae, and juvenile starfish still in the early stages of existence, prior to the hardening of their body walls. This approach addresses the issue of starfish at its source, effectively mitigating their population growth. At the same time, we have also considered whether the introduction of carnivorous fish will have an impact on the populations of marine species. When carnivorous fish are introduced to remove immature starfish from aquaculture areas, these fish may also eat small fish, crustaceans, and so on, and so assessment of the broader impact of this intervention is important.

## 3. Exploitation and Utilization of Starfish

The rampant proliferation of starfish poses a significant threat to both the marine aquaculture industry and the broader ecosystem. Addressing this challenge necessitates a dual-pronged strategy. Firstly, there is an urgent need to amplify research efforts related to the prevention and control of starfish outbreaks. Simultaneously, efforts should be channeled towards the development and comprehensive utilization of starfish resources, thereby transforming an environmental nuisance into a valuable asset.

Over the past few decades, researchers worldwide have conducted extensive research, laying the foundational knowledge and justification for the application of starfish in diverse fields, including evolutionary and developmental biology [45], comparative physiology [46,47,48,49], medicine [50,51,52,53], food production [54], and industry [55,56]. This approach aims to harness the potential of starfish as a valuable resource, thereby mitigating the adverse impact of their proliferation on marine ecosystems and creating opportunities for sustainable and innovative utilization. Exploitation and utilization of starfish are shown in Table 3.

### 3.1. Basic Biological Research

Starfish and other echinoderms are of special interest from an evolutionary perspective because they are deuterostomes, along with chordates (including vertebrates) and hemichordates, whereas the majority of animal phyla are protostomes. Therefore, research on starfish and other echinoderms can shed light on the evolutionary origins of vertebrate characteristics [45]. Furthermore, echinoderms are unique amongst bilaterian animals in having acquired pentaradial symmetry. Thus, during the life cycle of starfish and other echinoderms, the larval stage exhibits bilateral symmetry but then metamorphoses into a juvenile and then an adult that typically exhibit a pentaradial body plan. There is interest amongst zoologists and development biologists in understanding how this transformation occurred evolutionarily and occurs developmentally in extant echinoderms. Accordingly, using starfish as an experimental model, insights into this issue have been reported recently [45].

The phylogenetic position of echinoderms as invertebrate deuterostomes has provided a rationale for using starfish to investigate molecular evolution by providing an invertebrate ‘link’ between vertebrates and protostomes that are widely used as experimental models (e.g., the insect *Drosophila melanogaster* and the nematode *Caenorhabditis elegans*). An example of this approach is the identification of neuropeptides and their cognate receptors in the starfish *A. rubens*, which has shed light on the evolutionary origin and phylogenetic distribution of neuropeptide signaling systems [46,47,48,49]. Furthermore, the functional characterization of neuropeptide signaling systems in *A. rubens* and other starfish species has revealed physiological roles in the regulation of feeding behavior [61,62,63] and reproduction [13,64].

Another aspect of biology where starfish have yielded valuable insights is research on biological adhesives. The tube feet of starfish produce adhesive proteins that enable them to temporarily adhere to the substratum during locomotion or to prey during feeding. The characterization of one of these proteins, Sfp1, revealed that it is cleaved into four functional subunits, which have specific binding domains for proteins, carbohydrates, and metals [9]. Two of these subunits were artificially produced as recombinant fragments in *E. coli*. These fragments were observed to self-assemble and form aggregates in NaCl, with the ability to adsorb onto surfaces in the presence of Na⁺ and/or Ca^2+^ ions. These proteins may have promising applications in cell culture and biomedicine, demonstrating non-cytotoxic effects on HeLa cells [65]. More recently, a catalog of other starfish footprint proteins (Sfps) have been identified in *A. rubens*. By analyzing their cellular expression patterns and conserved functional domains, diverse functions for the identified Sfps have been proposed [66]. For example, a de-adhesive gland cell-specific astacin-like proteinase weakens the binding between the adhesive material and the tube foot surface during degumming. By elucidating the temporary adhesion mechanism of starfish tube feet, this research may provide the foundations for development of novel materials for underwater adhesion and biomedical applications.

### 3.2. Medical Applications

The marine environment is a treasure trove of natural bioactive compounds, with echinoderms contributing significantly to marine natural products, accounting for nearly one-third of such compounds [67]. Echinoderms produce a variety of bioactive compounds, including saponins, fatty acids, lipids, and peptides, highlighting their potential as a valuable resource for the development of innovative pharmaceuticals.

In the face of the growing challenge of antibiotic resistance, the search for new antibacterial agents from natural sources has become increasingly critical. Research conducted by Kimura et al. [57] on the antibacterial potential of *Luidia clathrata* revealed that an ethyl acetate extract of body wall tissue displayed significant inhibitory effects against all ten strains of both Gram-positive and Gram-negative bacteria tested. In contrast, gonadal extracts, also extracted with ethyl acetate, showed inhibitory activity against only six of the strains. Interestingly, the gonadal extract did not exhibit activity against *Pseudomonas aeruginosa*, *Enterococcus faecalis*, *Bacillus cereus*, and *Mycobacterium smegma*, and its overall antibacterial efficacy was comparatively lower than that of the body wall extract. More specifically, progress has been made recently in the characterization of antimicrobial peptides derived from starfish, including *A. amurensis* [50]. Furthermore, the structure of a cysteine-rich antimicrobial peptide from the starfish *P. pectinifera* (PrCrAMP) has been determined, revealing that PrCrAMP is the prototype for a family of related peptides in echinoderms [15]. The research underscores the potential of starfish-derived compounds in medical applications, particularly in the fight against antibiotic-resistant bacterial infections.

The medicinal potential of starfish is largely attributed to water-soluble saponins and the presence of polyhydroxysterols, with the stomach and pyloric caecum serving as the primary reservoirs of saponins. Significant progress has been made in the field of anticancer therapy through the development of starfish saponin-based formulations, which have shown considerable efficacy in clinical practice [51]. Research conducted by Wang revealed the significant inhibitory effect of total starfish saponins on transplanted sarcoma S180 in mice, highlighting their anticancer properties [52]. Additional experiments demonstrated that purified exogenous lectin extracted from starfish has a toxic effect on various tumor cells. Moreover, starfish polyhydroxysterols have the ability to inhibit Ca^2+^ channels in myocardial cell membranes, thereby slowing the inward flow of Ca^2+^ and showing significant potential in the treatment of arrhythmia [53].

An intriguing study involving starfish sterol included assessments of “Y-maze”, “platform jumping”, and “dark avoidance” in mice, which yielded results indicating improved memory function. Specifically, these findings were characterized by reduced maze exit times and fewer errors in platform and dark box tests, signifying the notable impact of starfish sterol on memory enhancement [68]. Furthermore, the extracts of echinoderms such as starfish and sea urchins have been used in the treatment of various medical conditions, including asthma, alcoholism, bronchitis, diabetes, and heart disease, particularly in regions like Brazil [69]. Eicosapentaenoic acid (EPA) and docosahexenoic acid (DHA), both highly prevalent in starfish gonads, offer unique therapeutic potential. EPA and DHA are widely recognized for their roles in reducing blood pressure, antibacterial properties, and anti-inflammatory effects [51]. Ji et al. [70] incorporated enzymatic extracts obtained from starfish gonads into the diet to assess the levels of superoxide dismutase (SOD) and malondialdehyde (MDA) in the brain, blood, liver, heart, spleen, and other organs of mice. The results of the study showed a significant increase in SOD content across all organs and a concurrent decrease in MDA content in the experimental group of mice compared to the Yang-deficiency-type mice. These changes were accompanied by a significant enhancement in the activity levels and overall vitality of the experimental group. This outcome underscores the ability of enzymatic hydrolysis extracts from starfish gonads to mitigate lipid peroxidation reactions induced by free radicals in animals, thus effectively alleviating symptoms associated with Yang deficiency.

Additionally, many echinoderms, with starfish being a prime example, exhibit remarkable regenerative abilities. A study on arm tip regeneration following amputation in *A. rubens* revealed the stages of arm tip regeneration and associated patterns of cell proliferation and neuropeptide expression [71]. Accordingly, scientists are actively researching the regenerative potential of echinoderm appendages, aiming to gain insights into their applicability for understanding and treating neurodegenerative diseases in humans [69].

### 3.3. Food Applications

Starfish represent a viable marine food source with nutritional value for human consumption [72]. The edible part of the starfish comprises the reproductive organs, including the testes and ovaries, collectively known as starfish gonads. These gonads are non-toxic and possess a subtly pungent flavor coupled with a delightful taste [54]. Notably, the gonads are abundant in fatty acids, particularly eicosapentaenoic acid (EPA) and docosahexaenoic acid (DHA), which are the predominant omega-3 polyunsaturated fatty acids, providing essential nutritional benefits. Table 4 offers a comparative analysis of the nutritional composition of three representative echinoderms: *A. amurensis* (gonad), *Mesocentrotus nudus* (gonad), and *A. japonicus* (body).

The data presented in Table 4 elucidate that sea urchins possess the highest EPA content, with comparable levels observed in starfish and sea cucumbers. Conversely, starfish have the highest proportion of DHA. The ratios of essential amino acids to total amino acids in starfish, sea urchins, and sea cucumbers are relatively consistent. Starfish have elevated protein and fat content compared to sea urchins and sea cucumbers, and notably, they maintain a lower cholesterol level, rendering them a suitable dietary option for individuals with cardiovascular concerns [58,73].

The flavor- and taste-contributing amino acids (FAAs) function as indicators of seafood freshness. In the gonads of *S. dawsoni*, FAAs constitute 51.16% of the total amino acid content, slightly below that of clams (*Mactra veneriformis*) but comparable to abalone [76], underscoring the flavorful and appealing nature of starfish gonads. Recognizing starfish as a delectable delicacy has the potential to transform waste into a valuable resource, benefiting fishermen and fostering increased enthusiasm for fishing, while concurrently mitigating potential challenges in breeding farms.

In coastal cities such as Qingdao City in China, the consumption of starfish *A. amurensis* at dining tables has risen significantly due to their increasing population. The delightful and cost-effective flavor of starfish gonads has gained popularity among diners. As a result, starfish dishes have become a prevalent option on the menus of seafood restaurants in the region. Starfish can be cooked in a variety of ways: the gonads can be steamed with eggs, offering a texture reminiscent of fish roe and a flavor akin to crab roe. Moreover, the gonads can be preserved in cans [58]. During the breeding season, the starfish yields a high content of starfish gonads, approximately 20–25% of its own weight. Preparing a box of canned food with starfish gonads, enhanced by seasonings, requires only 2–3 starfish, making it an affordable and economically appealing option.

While starfish are considered a delicacy in China, they are rarely found on dining tables in other countries. Their filter-feeding behavior, primarily on shellfish, raises concerns about potential heavy metal accumulation in their tissues. Danis et al. [77] conducted tests on heavy metal enrichment using *A. rubens* specimens collected from areas near the plume of the Scheldt river. Their findings revealed the presence of four types of heavy metals (Zn, Cd, Cu, and Pb) in the body wall and pyloric caeca. Notably, there was a significant concentration of Pb in the skeleton, contrasting with relatively lower levels in other tissues [78]. Furthermore, the gonads exhibited elevated concentrations of Zn and Cd, known to negatively impact gonadal development and potentially lead to the production of defective offspring [79]. It is noteworthy that there have been no reported cases of heavy metal enrichment in *A. amurensis* in China. Nevertheless, it would be prudent to exercise caution when consuming starfish, emphasizing the importance of recognizing their origin, species, and diet.

### 3.4. Industry and Other Applications

#### 3.4.1. Models as Lightweight Industrial Materials

In the realm of structural materials development, the imperative to reduce material weight for minimizing consumption, energy usage, and environmental impact remains paramount. A practical strategy involves creating a honeycomb-like structure by introducing pores. The starfish’s body wall, comprising calcite ossicles connected by collagenous tissue and muscle, exhibits variable stiffness attributed to the mechanical variability of the collagenous component [80,81].

A groundbreaking study by Blowes et al. [76] employed X-ray microtomography to observe the 3D structure of the endoskeleton of the European starfish *A. rubens*. This exploration revealed the shapes and interactions of various ossicle types, forming a mesh-like structure determining the starfish’s body shape variability. The arrangement of overlapping rhombic ossicles around the body wall skeleton’s empty spaces is functionally significant for gas exchange between coelomic fluid and external seawater. *Proforeaster nodosus*, prevalent in coral reef areas, features a distinctive and robust double-scale microcrystalline lattice structure, positioning it as an ideal model for structural materials [81].

The lightweight construction of starfish provides an exemplary model for the development of industrial materials. Studying the mechanical properties of echinoderm body wall tissue is invaluable in materials science, particularly for comprehending the in vivo formation mechanisms of these intricate and highly periodic microstructures.

#### 3.4.2. Application of Body Wall Collagen

The protein-rich body walls of starfish present an opportunity for collagen extraction, subsequently processed into gelatin [51,54]. Vate et al. [55] employed high-shear mechanical homogenization (HSMH) and ultrasound (US) technology to pretreat collagen extraction from starfish, reducing processing time without compromising collagen quality. Guo [56] isolated and purified collagen from *A. amurensis* body walls for biological function studies, demonstrating favorable outcomes with corneal cells, skin cells, and cultured tumor cells. The body wall of *P. pectinifera* is composed of 10% muscle proteins, with collagen constituting 60% of this composition [57]. Collagen, as the principal structural protein in the extracellular matrix of connective tissues, finds extensive applications in biotechnology, particularly in tissue regeneration [82].

Diversifying its applications, collagen can be hydrolyzed into collagen peptides through protein hydrolysis [83]. Characterized by low molecular weight and good water solubility, collagen peptides can easily penetrate deep skin layers, offering diverse functions such as wound healing, bone regeneration, and anti-aging effects [84,85,86]. Consequently, starfish emerge as promising raw materials for cosmetics [87].

#### 3.4.3. Feed Additives

Starfish can serve as a feed ingredient to enhance the growth of aquatic animals. Wu [59] demonstrated improved average weight gain and specific growth rates in *A. japonicus* by adding 20–30% of dried starfish powder to the feed, compared to the control group. Similarly, Xu [58] incorporated dried starfish powder into abalone feed, with experiments showing that adding 30% of dried starfish powder to domestic abalone feed produced feeding effects on par with imported bait. Therefore, incorporating dried starfish powder into animal feed presents promising prospects, including enhanced feed utilization and reduced waste accumulation, among other potential benefits.

#### 3.4.4. Contributing to the Reduction in Carbon Emissions

Starfish play a crucial role in the natural carbon cycle by absorbing carbon from seawater and utilizing it to form exoskeletons primarily composed of inorganic salts like calcium carbonate. Consequently, after their demise, a substantial portion of the carbon-containing materials within their bodies remains deposited on the seabed, contributing to the reduction in carbon emissions into the atmosphere from the ocean [87].

## 4. Conclusions

In conclusion, the occurrence of starfish outbreaks poses a substantial threat to local mariculture, arising from various sources. Key factors encompass the resilient survival capabilities of starfish, the abundant food resources provided by cultivated shellfish, and the declining population of natural predators, all contributing to their unchecked proliferation. Researchers from diverse countries have engaged in studies focusing on preventive and control measures, achieving significant strides in the exploration and utilization of biological resources. Nevertheless, tackling the underlying causes of global starfish outbreaks demands sustained efforts and advancements.

We contend that attributing starfish outbreaks solely to their robust survival abilities or significant feeding habits oversimplifies the issue. Starfish play vital roles in the marine benthic ecosystem. The underlying cause of starfish outbreaks can be traced to the imbalance of the marine ecosystem, resulting from the accumulation of various ecological crises over time. To effectively address global starfish outbreaks, specific measures should be implemented.

**Environmental Stewardship:** Address the core deterioration of the overall environment. Safeguard the ecological environment and restore equilibrium in coastal areas.

**Collaboration:** Foster collaborative efforts among fishermen, universities, and scientific research institutions to monitor changes in planktonic larvae abundance, adult starfish distribution, and their movement patterns. Analyze and interpret collected data to provide timely warnings, facilitating the development of well-informed strategies, species selection, specifications, and quantities to minimize losses and maximize income while ensuring a healthy culture.

**Research and Investigation:** Intensify research into the behavior and reproduction of starfish to mitigate potential hazards, protect the ecological environment, and enhance the economic prospects of shellfish aquaculture areas. Implementing these measures offers a strategic approach to addressing the challenges posed by starfish outbreaks and working towards the restoration of marine ecosystem balance, concurrently ensuring the sustainability of mariculture activities.

## Figures and Tables

**Figure 1 biology-13-00537-f001:**
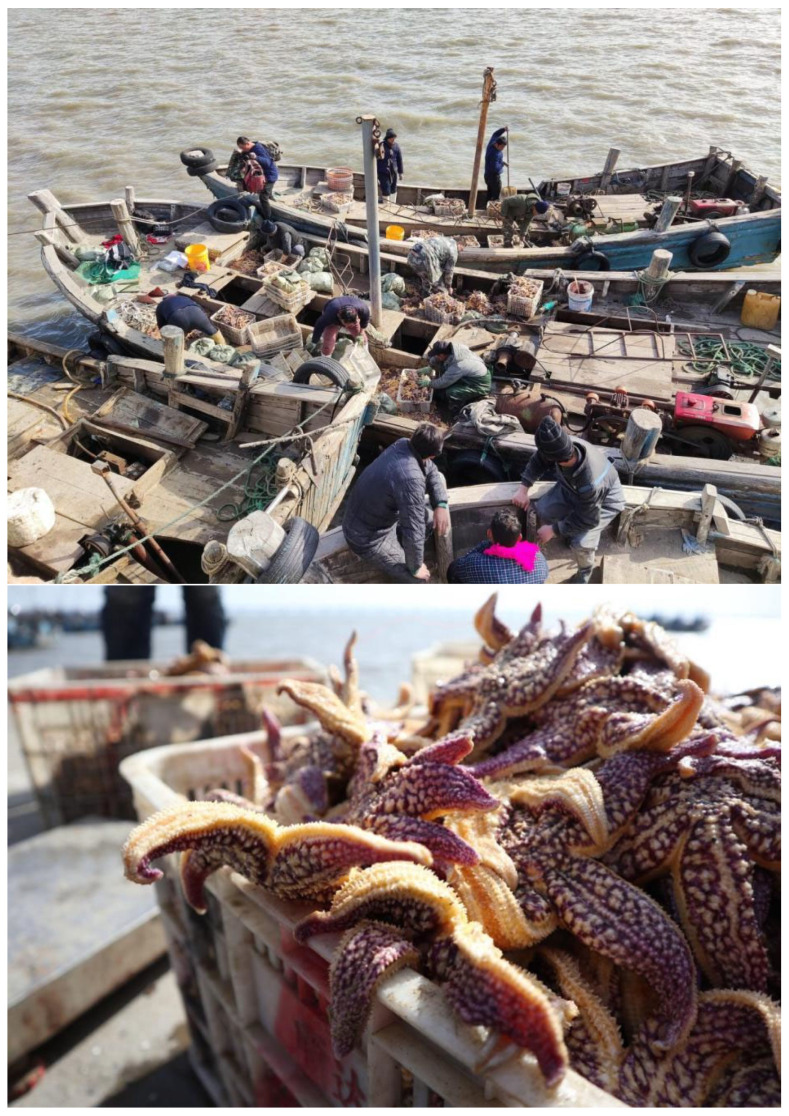
Starfish (*Asterias amurensis*) collected in March 2021, following an outbreak in Jiaozhou Bay, China (Image: Alamy) (https://chinadialogueocean.net/en/conservation/17894-whats-causing-plagues-of-seastars/ (accessed on 28 May 2024).

**Figure 2 biology-13-00537-f002:**
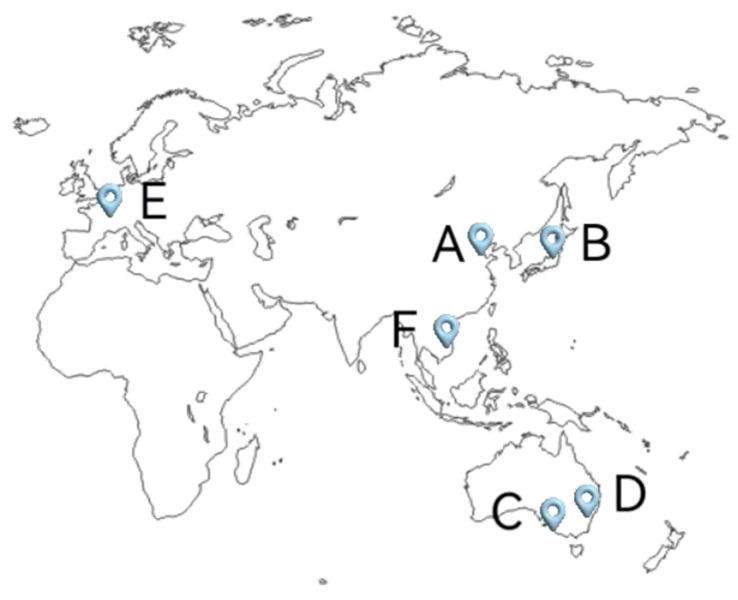
Map showing the locations of starfish outbreaks for different species in various regions of the world. A: *Asterias amurensis* and *Patiria pectinifera*; B: *Asterias amurensis*; C: *Acanthaster planci*; D: *Asterias amurensis*; E: *Asterias rubens*; F: *Acanthaster planci*.

**Table 1 biology-13-00537-t001:** Starfish outbreaks in various regions.

Time	Region	Species of Starfish	Endangered Species
1950s [14,15].	Tokyo Bay, Sendai Bay, and Yuming Bay in Japan	*Asterias amurensis*	Shellfish
1960s [24]; 1985–2012 [25].	Great Barrier Reef in Australia	*Acanthaster planci*	Coral
1974 [27].	Tottori Prefecture,	Starfish	*Babylonia japonica*
1981 [10].	Japan	*Asterias amurensis*	*Scapharca broughtoni*
1982–1988 [16].	Bay of Douarnenez, Brittany, France	*Asterias rubens*	Mussel beds
1980s [21].	Southern Australia	*Asterias amurensis*	Scallop larva
End of the 20th century [19,28].	Tasmania	*Asterias amurensis*	Mussels, *Brachionichthys hirsutus*
2006–2011 [29]; 2018 [30].	Paracel Islands in the South China Sea	*Acanthaster planci*	Coral
2006–2007 [6]; since 2020 [22].	Qingdao, Shandong	*Asterias amurensis*, *Patiria pectinifera*	Abalone, *Ruditapes philippinarum*, *Apostichopus japonicus*, etc.
2007 [8].	Qingliu River	*Asterias*	*Azumapecten farreri*
2016–2018 [31].	Nha Trang Bay, Vietnam	*Acanthaster planci*	Coral

**Table 2 biology-13-00537-t002:** Methods of starfish removal.

Control and Prevention Methods	Ways	Process	Chemical Components
Physical removal methods	Manual timed cleaning	Large-scale fishing based on the reproductive cycle of starfish.	
Chemical removal methods	Drug delivery		Quicklime; Ammonium salt and acetic acid stimulation; Acid-based Pharmaceuticals; Pheromones.
Removal methods through natural enemy deployment	Attack from predators	Destroying starfish with *Triton’s* *trumpet*. *Leptomithrax gaimadii*, *Larus argentatus*, fish, etc.	

**Table 3 biology-13-00537-t003:** Exploitation and utilization of starfish.

Exploitation and Utilization of Starfish	Ways	References
Basic biological	Research on starfish and other echinoderms can shed light on the evolutionary origins of vertebrate characteristics.	[45]
Medical applications	Characterization of antimicrobial peptides; Anticancer therapy through the development of starfish saponin-based formulations.	[50,51,52,57]
Food applications	The gonads of starfish are abundant in fatty acids, particularly eicosapentaenoic acid (EPA) and docosahexaenoic acid (DHA).	[54]
Industry applications	The protein-rich body walls of starfish present an opportunity for collagen extraction, subsequently processed into gelatin.	[51,54]
Other applications	Starfish can serve as a feed ingredient to enhance the growth of aquatic animals, contributing to the reduction in carbon emissions.	[58,59,60]

**Table 4 biology-13-00537-t004:** Comparison of nutritional components of three echinoderms.

Species	EPA/%	DHA/%	EAA/TAA/%	Protein/%	Crude Fat/%
*A. amurensis* [10,73].	10.35	9.19	37.32	16.60	11.40
*M. nudus* [74,75].	22.15	1.50	39.61	13.46	3.29
*A. japonicus* [75].	10.00	8.44	34.47	5.57	0.60

Note: EAA, essential amino acid; TAA, total amino acid.

## Data Availability

No new data were created or analyzed in this study. Data sharing is not applicable to this article.

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
