# Peer review of "Research Progress on Starfish Outbreaks and Their Prevention and Utilization: Lessons from Northern China"

_biology, 2024, doi:10.3390/biology13070537_

Round 1
Reviewer 1 Report
Comments and Suggestions for Authors
Plagiarism:
9%. OK
Title:
The title seems like, a bundle of research has already described the consequences of starfish outbreak in different bays, seas, oceans, etc. and those research only specifically focused on the starfish. While I saw the Table 1, its suggesting only few studies and there is no global pattern for it, such as prevalence in Asia, Europe, Americas, etc.
I think the author should rewrite the title of the manuscript and need to more focus and adjacent to the objective and conclusion of the review.
Abstract:
Too short. I want to see some digit, values and explanation rather just describe the conclusion of the synthesis.
Keywords:
Are you specifically focusing on a single species (Asterias amurensis?), if so, please reflect the terms in your manuscript title
Introduction:
-Please provide more background of the subject topics of the manuscript.
-Please write why this review writing was important, although we have a lot of topics, why you choose this topics?
-What was the motivation of the working on this topics?
Materials and methods:
This review could be a systematic synthesis of the previous result. I did not see any such approach.
-How did you conduct the document search on starfish disturbing shellfish?
-What process/methodology you applied to select a paper to be cited or not to be cited in your current review?
-How your data was collected and synthesized? I mean, the reader would like to know the process of your desktop data collection process.
2. Starfish outbreak disasters and their causes
L66-69: Where is the reference of this statement?
L61-83: All examples are provided here from China, then why you are claiming this review is a global?
Figure 1: You must declare the copyright of this image.
L106-107: What do you mean by extensive research? The citation you provided is suggested that it’s only in northern China
L109: Where is this hypothesis? I did not find it on [16]
L113: Where is this hypothesis? I did not find it on [32]. Please send me the paper in English translation, I need to check this hypothesis.
L136-139: This section must not be under this sub-heading. Please write a new sub-heading, "Ocean transportation". I think this section is not important as this section can be considered as invasive species introduction.
L148-149: If it’s a global review, why authors repeatedly used examples from a certain country?
L155: Please write the climate change effect in the title
L169-171: Are you concluding here? Why the conclusion is here. Are you using AI to write the paragraph?
2.3 sub-headings: Please prepare a list of control and prevention methods, way, process, and chemical components, etc. in a table.
2.3.3 Sub-heading: Is it a called biological control?
3. Exploitation and utilization of starfish
L279: Please provide a table of collected published researches and their outcome summery.
L286-292: These sentences require more citations.
L296-297: Please cite some researches.
L444-449: Requires citations.
4. Summery
Please write the conclusion part of the manuscript.
References:
32. Please write in which language this paper was published? Please write it anywhere in the reference
Please check the commented file attached.

Author Response
Title:
Comment 1:The title seems like, a bundle of research has already described the consequences of starfish outbreak in different bays, seas, oceans, etc. and those research only specifically focused on the starfish. While I saw the Table 1, its suggesting only few studies and there is no global pattern for it, such as prevalence in Asia, Europe, Americas, etc.
I think the author should rewrite the title of the manuscript and need to more focus and adjacent to the objective and conclusion of the review.
Response 1:Thank you for your suggestion. We’ve changed the title to “Research progress on starfish outbreaks and their prevention and utilization: lessons from northern China”.
Abstract:
Comment 2:Too short. I want to see some digit, values and explanation rather just describe the conclusion of the synthesis.
Response 2:We have modified this, see L30-32 in revised manuscript.
Keywords:
Comment 3:Are you specifically focusing on a single species (Asterias amurensis?), if so, please reflect the terms in your manuscript title
Response 3:We are not specifically focusing on a single species. In our review, we focus on multiple species, such as Asterias amurensis, Patiria pectinifera, Solaster dawsoni, Acanthaster planci, Asterias rubens, etc. See L44-45, L104, etc in revised manuscript.
Introduction:
Comment 4:-Please provide more background of the subject topics of the manuscript.
-Please write why this review writing was important, although we have a lot of topics, why you choose this topics?
-What was the motivation of the working on this topics?
Response 4:Thank you for your comment. We’ve provided more background of the subject topics of the manuscript. See L52-53 in revised manuscript.
Materials and methods:
Comment 5:This review could be a systematic synthesis of the previous result. I did not see any such approach.
-How did you conduct the document search on starfish disturbing shellfish?
-What process/methodology you applied to select a paper to be cited or not to be cited in your current review?
-How your data was collected and synthesized? I mean, the reader would like to know the process of your desktop data collection process.
Response 5:This review collected and sorted out a number of domestic and foreign references on starfish from the academic literature available online, including the hazards of starfish outbreak and related management measures, and from which we put forward some of our own opinions.
- Starfish outbreak disasters and their causes
Comment 6:L66-69: Where is the reference of this statement?
Response 6:Reference to [6].
Comment 7:L61-83: All examples are provided here from China, then why you are claiming this review is a global?
Response 7:We’re sorry that we have caused this misunderstanding. The title and the introduction have been revised to make it clear that this review mainly introduced starfish disasters occurring in China, but not the whole world.
Comment 8:Figure 1: You must declare the copyright of this image.
Response 8:We consider these images to be public news and we have credited the official website of the images.
Comment 9:Please provide a map of this reported loation. Use the picture of the species on map.
Response 9:We have provided a map. See L111-115.
Comment 10:L106-107: What do you mean by extensive research? The citation you provided is suggested that it’s only in northern China
Response 10:We have modified this, see L118 in revised manuscript.
Comment 11:L109: Where is this hypothesis? I did not find it on [16]
Response 11:The references [16] here were mislabeled, and this hypothesis comes from reference [32], and we have modified it.
Comment 12:L113: Where is this hypothesis? I did not find it on [32]. Please send me the paper in English translation, I need to check this hypothesis.
Response 12:This is a Chinese reference, We can translate the "hypothesis" paragraph for you: There are two hypotheses about the cause of starfish disaster outbreak, one is "Floating food growth hypothesis", and another is "the hypothesis of decreasing number of natural enemies".
Comment 13:L136-139: This section must not be under this sub-heading. Please write a new sub-heading, "Ocean transportation". I think this section is not important as this section can be considered as invasive species introduction.
Response 13:This paragraph have been deleted.
Comment 14:L148-149: If it’s a global review, why authors repeatedly used examples from a certain country?
Response 14:We have revised the title to make it clear that this review is not global.
Comment 15:L155: Please write the climate change effect in the title
Response 15:We have modified this, see L158 in revised manuscript.
Comment 16:L169-171: Are you concluding here? Why the conclusion is here. Are you using AI to write the paragraph?
Response 16:This is a summary of 2.2, and no we did not use AI. See L171 in revised manuscript.
Comment 17:2.3 sub-headings: Please prepare a list of control and prevention methods, way, process, and chemical components, etc. in a table.
Response17 :We have prepared a list, see L181 in revised manuscript.
2.3.1. Physical removal methods
Comment 18:2.3.3 Sub-heading: Is it a called biological control?
Response 18:Yes. Please see L261 in revised manuscript.
- Exploitation and utilization of starfish
Comment 19:L279: Please provide a table of collected published researches and their outcome summery.
Response 19:We have modified this, please see L 298 in revised manuscript.
Comment 20:L286-292: These sentences require more citations.
Response 20:We have modified this, please see L 293-294 in revised manuscript.
Comment 21:L296-297: Please cite some researches.
Response 21:We have modified this, please see L 303 in revised manuscript.
Comment 22:L444-449: Requires citations.
Response 22:We have modified this, please see L 445 in revised manuscript.
- Summery
Comment 23:Please write the conclusion part of the manuscript.
Response 23:We have renamed this paragraph as "Conclusion", please see L489 in revised manuscript.
References:
Comment 24:32. Please write in which language this paper was published? Please write it anywhere in the reference
Response 24:This paper was published in Chinese, and we have indicated it. Please see L602 in revised manuscript.
Comment 25:Please check the commented file attached.
Response 25:We have checked the commented file attached.
Reviewer 2 Report
Comments and Suggestions for Authors
2.3.1. Physical removal methods
lines 184 and 185 --> The physical removal of the starfish does not have any impact on the benthic flora and fauna? How is this removal performed? What are the fishing gears used? If they catch starfish, does it mean that is not selective?
It explains the consequences of chemical techniques for starfish removal. Still, it is unclear if the physical removal (by fishing boats) affects the environment where the starfish are found.
2.3.3. Removal methods through natural enemy deployment --> Similar question as the one above, it is understood how this method will help with the removal of some species of starfish, but won't these natural enemies have an impact over other marine species populations?
Author Response
Comments 1:Lines 184 and 185 --> The physical removal of the starfish does not have any impact on the benthic flora and fauna? How is this removal performed? What are the fishing gears used? If they catch starfish, does it mean that is not selective?
It explains the consequences of chemical techniques for starfish removal. Still, it is unclear if the physical removal (by fishing boats) affects the environment where the starfish are found.
Response :This physical removal method relies mainly on fishers casting nets to drag the starfish. This method may also catch a small amount of seaweed and benthic shellfish, but most are empty shells that have been eaten by the starfish.
Comments 2:Similar question as the one above, it is understood how this method will help with the removal of some species of starfish, but won't these natural enemies have an impact over other marine species populations?
Response :We thought about that as well. In China, we introduce carnivorous fish in aquaculture region to remove immature starfish. These fish may also eat small fish, shrimp, insects, etc., but this is the result of natural selection and we think it will not cause ecological damage.